# Spatial Heterogeneity of Watershed Ecosystem Health and Identification of Its Influencing Factors in a Mountain–Hill–Plain Region, Henan Province, China

Hejie Wei [1], Qing Han [1], Yi Yang [2,*], Ling Li [1,3] and Mengxue Liu [4]

1 College of Resources and Environmental Sciences, Henan Agricultural University, Zhengzhou 450046, China; hjwei@henau.edu.cn (H.W.); hanqing68@stu.henau.edu.cn (Q.H.); liling@henau.edu.cn (L.L.)
2 College of Forestry, Henan Agricultural University, Zhengzhou 450046, China
3 Henan Engineering Research Center of Land Consolidation and Ecological Restoration, Henan Agricultural University, Zhengzhou 450046, China
4 Faculty of Geographical Science, Beijing Normal University, Beijing 100875, China; mengxueliu@mail.bnu.edu.cn
* Correspondence: yangyi023@henau.edu.cn

**Abstract:** A watershed ecosystem is a compound ecosystem composed of land and rivers, and its health is closely related to the sustainable development of the region it is located in. The Yihe River Basin (YRB) in central China's Henan province, which is located in the north–south transition zone and has a mountain–hill–plain landscape from the upstream to the downstream, is adopted as the research area in this study. A watershed ecosystem health assessment system is constructed based on an ecosystem vigor–organization–resilience–service supply and demand harmony (EVORSH) framework and utilized to assess the ecosystem health in the YRB by taking a 3 km × 3 km grid as the evaluation unit. Thirteen factors are selected from natural and human social factors, and from them, the factors that influence watershed ecosystem health through the generation of spatial heterogeneity are identified using the geographical detector model. The following findings are obtained. (1) The mean value of ecosystem health levels in the YRB is 0.65 and at the good level. The ecosystem health has considerable spatial heterogeneity. The areas with high–high concentration are distributed in the mountains in the upper reaches of the YRB, and the areas with low–low concentration are mainly distributed in the plain areas in the middle reaches of the YRB. (2) The geographical detector result shows that 9 of 13 factors have a considerable impact on the spatial distribution of the YRB's ecosystem health. The interaction between two factors is enhanced synergically. The decisive power of population density, rainfall, and potential evapotranspiration are more than 0.5, so these three are the main factors that influence the distribution of ecosystem health in the YRB. (3) The EVORSH framework is suitable for the measurement of ecosystem health in the YRB. The evaluation result is consistent with the actual situation in the YRB. A 3 km × 3 km grid is used as the basic research unit, and it can more accurately and scientifically express the spatial heterogeneity of ecosystem health in the YRB compared with the macro evaluation unit. This study can provide a scientific basis for ecological protection and high-quality development planning in the YRB. By integrating multi-dimensional data and methods, the EVORSH framework proposed in this study can quickly and scientifically assess the status of ecosystem health, identify the influencing factors of spatial heterogeneity, and could be applied in other similar watersheds.

**Keywords:** ecosystem health; ecosystem service supply and demand; geographical detector; Yihe River Basin

## 1. Introduction

With the rapid development of the economy and rapid growth of the population worldwide, physical resources in the ecological environment are being constantly consumed, which exerts enormous pressure on the ecological environment [1,2]. The problems

in the ecological environment and ecological security have become increasingly prominent, and ecosystem health is decreasing [3]. People are paying increased attention to the advantages and disadvantages of the ecological environment because of worsening environmental problems. Since the issuance of the sustainable development strategy policy [4], the concept of green development has been eliciting support among people [5,6]. People devote extensive attention to environmental protection and the rational development and application of resources. A healthy ecosystem is the core guarantee for achieving sustainable human development. The natural resources needed in the life activities of mankind should be provided by a healthy ecological environment [7]. As an important part of the evaluation of the regional landscape, ecosystem health assessment is an embodiment of the regional ecological environment, human activities, and socio-economic intensity, and it is a crucial assessment direction of sustainable development strategies [8,9]. Ecosystem health assessment can objectively evaluate the advantages and disadvantages of the ecological environment and analyze the main regional environmental problems and driving factors, thus providing new ideas for the comprehensive management of the regional environment [10]. The assessment of ecosystem health and the identification of regions with severe ecological problems are conducive to the scientific management and protection of the ecological environment. In 1941, the concept of land was linked to the notion of health [11]. Then, in the late 1980s, Rapport et al. put forward the definition of ecosystem health, that is, the state, condition, or performance of an ecosystem defined by practical standards, including primary productivity, nutrients, species diversity, instability, disease prevalence, size spectrum, and contaminants [12,13]. They systematically proposed three indicators for assessing ecosystem health: vigor, organization, and resilience. In the past, scholars defined ecosystem health on the basis of the health of the ecosystem itself and believed that if an ecosystem has the ability to recover under external threats, then it is healthy [14,15]. Some scholars have reported that research on ecosystem health should start from the ecosystem itself and consider the ability of the ecosystem to provide services for human beings [16,17]. Costanza et al. [18] believed that ecosystem health is closely related to human activities. On the basis of ensuring structural and functional integrity, ecosystem health should be integrated into ecosystem services to provide stable and sustainable services to human beings [19]. In short, ecosystem health is a comprehensive concept that provides a scientific basis for regional sustainable development and management.

From the perspective of assessment methods, ecosystem health is mainly assessed using the species indicator method and the indicator system method [20]. Aiming at a single ecosystem, the species indicator method mainly describes the health status of the ecosystem based on the number, diversity, and physiological indicators of the key indicator groups in the ecosystem [19,21,22]. Comprehensive ecosystem health is mainly assessed by the indicator system method. Compared with the indicator species method, the indicator system method integrates multiple indicators of the ecosystem to reflect the process of the ecosystem, highlights the evolution of ecological health related to humans, and accurately reflects the load and recovery capacities of the ecosystem after being threatened [19,23]. Common models for assessing ecosystem health by using the indicator system method include energy analysis [24], ecosystem vigor–organization–resilience (EVOR) [18,25–27], vigor–organization–resilience–service–function [28,29], pressure–state–response (PSR) [30–34], driving force–pressure–state–exposure–effect–action [35], and development level-service function-resistance of disturbance–maintenance models [36]. The PSR model was constructed by the Organization for Economic Cooperation and Development and is mainly used to reflect the causal relationship of the interaction between human beings and the ecosystem from the perspectives of pressure, status, and response [37–39]. However, when building the indicator system, the PSR model mostly relies on a subjective selection of assessment indicators, and ensuring the scientific nature of indicators is difficult [40].

The EVOR model proposed by Costanza et al. [18] was identified as a diagnostic indicator of ecosystem health by the International Conference on Ecosystem Health in 1999. In the EVOR model, the ecosystem health assessment system is comprehensively constructed

from vigor, organization, and resilience, and the model can fully reflect the health status of each subsystem in the regional ecosystem and objectively evaluate the comprehensive health status of the regional ecosystem [41,42]. The EVOR model rapidly constructs the assessment indicators on the basis of remote sensing image data to make an assessment [1]. This assessment model has been widely used because it is reasonable, effective, and simple [43–45]. For example, Ou et al. [46] dynamically evaluated the spatiotemporal evolution characteristics of ecosystem health in the Yangtze River Delta by using the EVOR model and found that the overall ecosystem health showed a decreasing trend during the two decades of 1995–2015, which better reflected the ecosystem health characteristics of the region. The health status of various ecosystems, such as wetlands [47], grasslands [48], watersheds [49,50], and cities [19,51], can also be assessed through the EVOR model. The traditional EVOR model pays particular attention to the integrity and sustainability of the natural ecosystem itself but ignores human factors [42]. It cannot connect the natural ecosystem with human social activities. With the deepening of related research, the EVOR model has been continuously expanded; examples include the combination of EVOR and ecosystem service models and the vigor–organization–resilience–contribution model [19]. In 2012, Costanza reported that a healthy ecosystem needs to be able to provide a series of valuable ecosystem services in a sustainable manner [52]. Therefore, ecosystem services are included in the traditional regional ecosystem health assessment. An ecosystem vigor–organization–resilience–service (EVORS) ecosystem health assessment framework has also been established. This model has clear measurement standards and sufficient ecological information for assessing ecosystem health. At present, it is widely accepted and constantly improved in relevant research [53]. For example, in consideration of the needs of human society, Pan et al. [51] proposed an EVORS supply and demand assessment framework to further extend the EVORS model. However, the expanded ecosystem health assessment model is not well-rounded in the selection of ecosystem service supply and demand indicators, and it lacks the selection of cultural ecosystem service; hence, it cannot accurately reflect the comprehensive service level of the ecosystem services.

Based on the previous research on ecosystem health, two knowledge gaps still exist in current research. First, scholars cannot reach a consensus regarding the subjectivity and scientific nature of the selection and number of indicators in the research process with the ecosystem health model. The selection of indicators is not well-rounded enough, and the indicator embodiment on the interaction between humans and nature is relatively weak. The ecosystem health assessment model can still be improved in terms of the index system composition. Further exploration is needed to find appropriate evaluation indicators. Second, most ecosystem health assessment frameworks focus on the measurement of the ecosystem health level. The factor interpretation and spatial differentiation mechanism of ecosystem health assessment results are still unclear.

Current ecosystem health research involves global, regional, landscape, ecosystem, and other levels. In terms of the assessment unit, urban agglomerations and administrative regions are adopted as analysis units in most studies [54–57]. In a few studies, spatial grids with high accuracy are employed for analysis [58]. In regions with few administrative units, the spatial grid, which can effectively highlight the micro level, is used as the assessment unit to evaluate ecosystem health. It is conducive to improving the refinement, universality, and credibility of the assessment results [59]. Therefore, the current study assesses the ecosystem health of the Yihe River Basin (YRB) by adopting the spatial grid, which can effectively embody spatial heterogeneity, as the assessment unit. The YRB, one of the branches of the Luohe River in the Yellow River Basin, is a composite system composed of terrestrial and freshwater ecosystems. It is located in the ecological transition zone and ecological environment sensitive zone in China. Research on watershed ecosystem health has focused on the Yangtze River Basin [19,46,49,51,56,60–64] in South China, but research on ecosystem health assessments of the Yellow River Basin in North China is limited. Assessing its ecosystem health is essential. In recent years, with the rapid development of the social economy and the acceleration of urbanization in the basin, large amounts of

resources have been used unreasonably, resulting in a series of ecological environment problems, such as vegetation damage and water pollution [65]. Moreover, the increasingly prominent mismatch between the supply and demand for ecosystem services has led to the deterioration of ecosystem health. In addition, Song County and Luanchuan County in the upper reaches of the basin are state-level, poverty-stricken counties. Therefore, the complex ecological and socio-economic problems in the YRB are intertwined and threaten local sustainable development [66]. Focusing on the construction of the ecosystem health framework by considering the mismatch between ecosystem service supply and demand, this study aims to perform an ecosystem health assessment of the YRB and explore the main factors that influence its distribution by adopting the spatial grid as the assessment unit. This study can provide a theoretical basis for the formulation of ecological protection policies in the YRB and for the improvement of human welfare.

## 2. Materials and Methods

### 2.1. Overview of the Study Area

The Yihe River is one of the important branches of the Luohe River on the south bank of the Yellow River Basin. Its geographical coordinates are N33°39′–34°41′, E111°19′–112°54′. It originates from the south bank of Xionger Mountain and passes through the north of Funiu Mountain. The river flows through Luanchuan County, Song County, Yichuan County, Luolong District, Yanshi City of Luoyang, and finally into the Luohe River. Its total length is about 264 km. The area of the basin is about 6100 km². The annual average temperature and annual rainfall are about 13–15 °C and 600 mm, respectively. A mountain–hill–plain landscape pattern is formed from the southwest to the northeast [66]. The research area has diversified species and rich ecological resources, and the terrain exhibits strong spatial heterogeneity. Conducting an ecosystem health assessment by adopting the YRB as the research area is representative and crucial for the ecological protection and high-quality development of the Yellow River Basin (Figure 1).

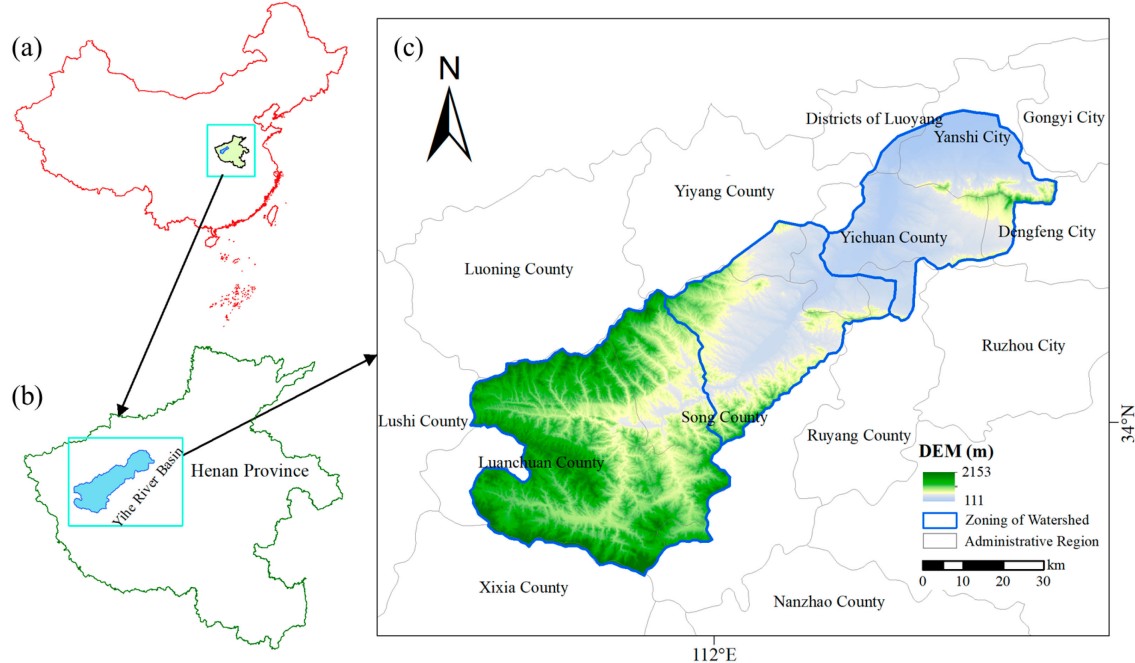

**Figure 1.** Location map of Yihe River Basin. (**a**) China; (**b**) Henan Province; (**c**) Yihe River Basin.

### 2.2. Data Sources

The data required for this study include basin land use/land cover change (LULCC) remote sensing monitoring data with a 30 m spatial resolution, digital elevation model (DEM) data, meteorological data, normalized difference vegetation index (NDVI) data, and

socio-economic data (including population, GDP raster data, and density of road networks) from 2018 or 2019. The sources are shown in Table 1. Land use classification data are provided by the Data Center for Resources and Environmental Sciences, Chinese Academy of Sciences. China's Multi-period LULCC Remote Sensing Monitoring Dataset provides relevant data with a spatial resolution of 30 m and an interpretation accuracy of 90%; the dataset is constructed through manual visual interpretation by the Chinese Academy of Sciences by taking US Landsat remote sensing data as the main source.

**Table 1.** Data sources.

| Data Name | Data Layout | Data Sources | Data Usage |
|---|---|---|---|
| Land use/land cover change (LULCC) remote sensing monitoring data | Raster data with resolution of 30 m | Resource and Environment Science and Data Center of the Chinese Academy of Sciences (http://www.resdc.cn/, accessed on 10 July 2022) | Basic data for calculating ecosystem services, cropland proportion, water coverage, and urbanization rate |
| Digital elevation model (DEM) data | Raster data with resolution of 30 m | The data are from the geospatial data cloud (http://www.gscloud.cn/, accessed on 10 July 2022) | Basic data for calculating ecosystem services, watershed elevation, slope, and slope direction |
| Temperature and rainfall data | Raster data with resolution of 30 m | China Meteorological Data Service Center (http://data.cma.cn/, accessed on 1 June 2020) | The factors of watershed ecosystem health |
| Normalized Difference Vegetation Index (NDVI) with 30 m resolution | Raster data with resolution of 30 m | Earth Big Data Science Engineering (CASEarth) Databank (http://databank.casearth.cn/, accessed on 1 June 2020) | The factor of watershed ecosystem health |
| Potential evapotranspiration | Raster data with resolution of 30 m | Resource and Environment Science and Data Center of the Chinese Academy of Sciences (http://www.resdc.cn/, accessed on 10 February 2023) | The factor of watershed ecosystem health |
| Gross domestic product (GDP) | Raster data with resolution of 1 km | Resource and Environment Science and Data Center of the Chinese Academy of Sciences (http://www.resdc.cn/, accessed on 10 July 2022) | The factor of watershed ecosystem health |
| Road network density | Vector data | https://www.openstreetmap.org, accessed on 10 July 2022 | The factor of watershed ecosystem health |

### 2.3. Research Framework

The ecosystem vigor–organization–resilience–service supply and demand harmony (EVORSH) framework of this study is based on the EVORS framework proposed by Constanza [18]. The enrichment and extension of the original framework are mainly reflected in two aspects. First, the connotations of ecosystem service and the indicator system are enriched. The harmonious index of the supply and demand of watershed ecosystem services is measured based on the supply and demand of ecosystem services, which reflects the pressure of human society on ecosystem health. Second, after assessing watershed ecosystem health, a local spatial autocorrelation analysis in GeoDa software is performed to identify the spatial pattern of watershed ecosystem health. The geographical detector model which was programmed in Excel is used to identify the factors that influence the spatial distribution of watershed ecosystem health. This study evaluates the health level of watershed ecosystems on the basis of traditional EVOR and EVORS frameworks, whose results are compared with the evaluation result of this study. In the process of index selection, an index system reflecting the ecosystem health in the YRB under its natural state is constructed. The selected indicators include net primary production (vigor indicator), landscape heterogeneity and landscape connectivity (organization indicator), and resistance and resilience coefficients of the ecosystem (resilience indicator) [18]. The

indicators of ecosystem services include the indicators related to provisioning, regulating, supporting, and cultural services. ArcGIS10.2, FRAGSTATS4.2, and Excel2016 software are used to process the multi-source data. Ecosystem vigor, ecosystem organization, ecosystem resilience, and ecosystem service supply and demand are calculated. Then, the ecosystem health index is obtained by the geometric mean [18,42,51]. Visualization function display and spatial analysis are performed for the ecosystem health status of the YRB by using ArcGIS software based on the spatial grid. The geographical detector model is adopted to explore the driving factors that affect the distribution. Afterward, the study's results on watershed ecosystem health and its influencing factors are discussed (Figure 2).

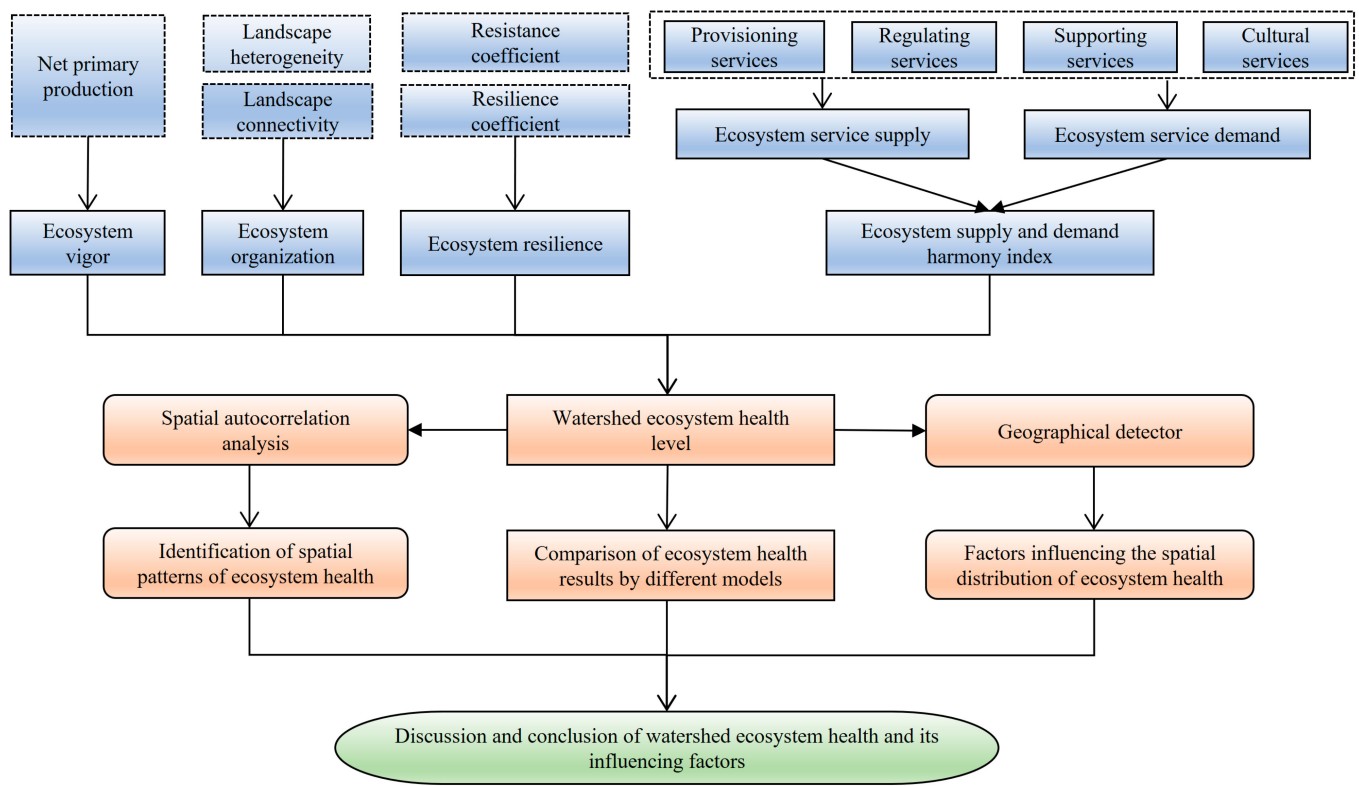

**Figure 2.** Research framework for watershed ecosystem health assessment and identification of the influencing factors.

### 2.4. Ecosystem Health Assessment

This study measures the ecosystem health index from four dimensions, namely, ecosystem vigor, organization, resilience, and harmony between the supply and demand of ecosystem services. To comprehensively study the spatial characteristics of ecosystem health in the YRB, a grid is selected as the research unit in this work. The commonly used grid units in the previous literature [67–74] include 500 m × 500 m, 1 km × 1 km, 3 km × 3 km, 5 km × 5 km, and 10 km × 10 km. In consideration of the impact of the minimum plasticity unit on the research results, the characteristics and differences of YRB's health pattern under different grid scales are compared. The results show that the 3 km grid scale analysis results are generally consistent and can express the spatial differences of the YRB. Hence, the 3 km × 3 km grid is adopted as the basic research unit in this study. Grid samples are created, and a total of 756 grids are obtained in the study area. The ecosystem health level of each grid is calculated separately. The formula is as follows [18,51]:

$$EHI_i = \sqrt[4]{V_i \cdot O_i \cdot R_i \cdot ESDH_i} \tag{1}$$

where $EH_i$ refers to the ecosystem health value of the $i$ unit, $V_i$ refers to the ecosystem vigor of the $i$ unit, $O_i$ refers to the ecosystem organization power of the $i$ unit, $R_i$ refers to the

ecosystem resilience of the $i$ unit, and $ESDH_i$ refers to the supply and demand harmony index ecosystem services in the $i$ unit.

Ecosystem health is a relative concept without a fixed standard. Referring to the classification standard for ecosystem health levels proposed by experts, this study divides the ecosystem health subsystems and ecosystem health indices are classified by the equivalent method in the YRB. The results are divided into five grades, namely, severe, poor, moderate, good, and excellent, with an interval of 0.2 [71,75,76] (Table 2).

**Table 2.** Reference for ecosystem health assessment grade.

| Health Status | Ecosystem Characteristics | Scoring Range |
|---|---|---|
| Severe | The ecosystem, with an extremely unreasonable structure, suffers from external interference seriously. It cannot be kept stable, and it easily deteriorates. The ecosystem has a poor ability to resist interference and maintain stability, with unsustainable development. | 0.0–0.2 |
| Poor | The ecosystem, with an unreasonable structure, suffers from external interference more seriously. It can be kept stable in the short term. The ecosystem has a weak ability to resist interference and maintain stability, with weak development sustainability. | 0.2–0.4 |
| Moderate | The ecosystem has a general structure and is kept stable. The ecosystem has a general ability to resist interference and maintain stability, with weak development sustainability. | 0.4–0.6 |
| Good | The ecosystem has a reasonable structure and suffers from controllable external interference. It can be kept stable in the long term. The ecosystem has a strong ability to resist interference and maintain stability, with better development sustainability. | 0.6–0.8 |
| Excellent | The ecosystem has a reasonable structure and suffers from controllable external interference. It can be kept stable in the long term. The ecosystem has a very strong ability to resist interference and maintain stability, with good development sustainability. | 0.8–1.0 |

The standardization formula for ecosystem vigor, ecosystem organization, ecosystem resilience, ecosystem service supply and demand harmony, and ecosystem health indices is as follows:

$$x_s = \frac{x_i - x_{min}}{x_{max} - x_{min}} \tag{2}$$

where $x_s$ refers to the standardization value of the data, $x_i$ refers to the $i$th index value, $x_{min}$ refers to the minimum value of this index, and $x_{max}$ refers to the maximum value of this index.

### 2.4.1. Ecosystem Vigor

Ecosystem vigor is used to describe the metabolism or primary production of the ecosystem. According to previous studies [51], net primary production (NPP) is an effective indicator of characterizing ecosystem vigor. In this study, NPP is used to evaluate ecosystem vigor in the study area. The calculation of NPP in the YRB is performed by referring to the calculation of Meng et al. [77] and Zhu et al. [78]. In the process of NPP, photosynthetically active radiation (PAR) and actual light energy utilization are key variables. PAR is calculated by the total solar radiation and the fraction of PAR absorbed by the vegetation canopy. The actual light energy utilization is calculated by growth-limited factors, including temperature and moisture stress on plant photosynthesis.

### 2.4.2. Ecosystem Organization

Ecosystem organization refers to the stability of the ecosystem structure. Ecosystem organization is mainly characterized by landscape heterogeneity and landscape connectivity. Landscape heterogeneity is represented by the Shannon evenness index (SHEI) and mean patch fractal dimension (MPFD). Landscape connectivity is represented by the connectivity between the whole landscape and patches with important ecological functions. Overall landscape connectivity is represented by the division index and the connectance index [79]. In addition, the water bodies and forest patches in the study area have a considerable ecological function in carbon storage, soil conservation, and biodiversity maintenance.

The corresponding connectivity is measured by the landscape shape index (LSI) and the connectance index. In terms of weight settings, in accordance with the literature, the overall landscape heterogeneity and landscape connectivity play equally important roles in ecosystem health [80], so the weight is set to 0.35. The weight of water bodies and forest patches is set to 0.15.

$$O = 0.35LH + 0.35EL + 0.30IPC = (0.20SHEI + 0.15MPFD)$$
$$+(0.20CONNECT_1 + 0.15DI)+ \qquad (3)$$
$$(0.10CONNECT_2 + 0.05LSI_1 + 0.10CONNECT_3 + 0.05LSI_2)$$

where $O$ refers to ecosystem organization; $LH$ refers to landscape heterogeneity; $MPFD$ and $SHEI$ refer to the average patch fractal dimension index and the Shannon evenness index, respectively; $ELC$ refers to entire landscape connectivity; $DI$ and $CONNECT_1$ refer to the landscape division index and the connectivity index, respectively; $IPC$ refers to the connectivity of patches (i.e., water bodies and forests) with important ecological functions; and $LSI_1$, $LSI_2$, $CONNECT_2$, and $CONNECT_3$ refer to the landscape shape index and connectivity index of water bodies and forests.

### 2.4.3. Ecosystem Resilience

Ecosystem resilience refers to the resistance of the regional ecosystem structure and function to natural disasters and human disturbance or the ability of the ecosystem to recover to its original state after being damaged. Ecosystem resilience can be measured by resistance and elasticity coefficients [56,81]. Elasticity is the ability to restore the original state of the ecosystem after serious damage. Different land use types play different roles in ecosystem restoration. A natural ecosystem can easily recover or resist external interference, whereas an artificial ecosystem is vulnerable to external interference [56,79]. By referring to the relevant literature [56,79,81], this study obtains the elasticity and resistance coefficients of the ecosystem under different land use types (Table 3).

**Table 3.** Elasticity coefficients of the ecosystem under different land use types.

| Land Use Types | Cultivated Land | Woodland | Grassland | Water | Unused Land | Construction Land |
|---|---|---|---|---|---|---|
| Elasticity coefficient | 0.40 | 0.50 | 0.80 | 0.70 | 1.00 | 0.20 |
| Resistance coefficient | 0.50 | 1.00 | 0.70 | 0.80 | 0.20 | 0.30 |
| R | 0.47 | 0.85 | 0.73 | 0.77 | 0.44 | 0.27 |

Regional development is slow and human interference is relatively low in the YRB. The weight of ecosystem resistance set in this study should be higher than the weight of resilience (0.7 and 0.3, respectively) [56]. In other words, for various land use types in the YRB, the elasticity coefficient is equal to 30% of the resilience coefficient plus 70% of the resistance coefficient. The relevant equation is as follows:

$$R = 0.7C_{resistan} + 0.3C_{elastic} \qquad (4)$$

where $R$ refers to ecosystem resilience and $C_{resistan}$ and $C_{elastic}$ are the resistance and elasticity coefficients of the ecosystem of land use type in the specific region, respectively.

### 2.4.4. Supply–Demand Harmony Index of Ecosystem Services

According to basic data and local natural and socio-economic characteristics, five representative ecosystem services in the basin, namely, water yield, food production, carbon sequestration, soil conservation, and cultural services (i.e., value of entertainment aesthetics, leisure eco-tourism, knowledge system, and natural heritage), are selected (Table 4).

**Table 4.** Reasons for selecting the ecosystem service indicators.

| Ecosystem Services | Selection Reasons |
|---|---|
| Food production (provisioning services) | The land in the lower reaches is fertile, mainly valley plains, and it is the main grain production area in the YRB. |
| Water yield (regulating services) | Water is essential for maintaining ecosystem functions. |
| Carbon sequestration (regulating services) | Carbon sequestration services play an important role in regulating climate and protecting the global ecological environment. |
| Soil conservation (supporting services) | There is a mountain canyon area in the upper reaches, with rock stratum joints developed and strong weathering influence. There is a hilly area in the middle reaches, with rich mineral resources, sparse vegetation, and serious water and soil losses. |
| Cultural services | Cultural service is also an important ecosystem service for meeting people's spiritual pursuits. |

The quantification methods of food production, water yield, carbon sequestration, soil conservation, and cultural services supply and demand are listed in Table 5. The calculation details of food production, water yield, carbon sequestration, soil conservation, and supply and demand can be found in Meng et al. [77]. The calculation details of cultural service supply and demand can be found in Meng et al. [66].

**Table 5.** Quantification of ecosystem service supply.

| Ecosystem Service Indicators | Quantification Methods of Ecosystem Service Supply | Quantification Methods of Ecosystem Service Demand |
|---|---|---|
| Food production | Based on the significant linear correlation between NDVI and food production, the statistical data on grain, meat, milk, and water products in the yield are spatialized. | Based on the population density, the per capita food demand is estimated by using the lowest value of the ideal range of per capita daily food consumption. |
| Water yield | Based on the water balance principle, the water yield of each grid is quantized by subtracting the actual evaporation from rainfall. | The demand for water production services is obtained by calculating the comprehensive water consumption per capita and raster data of population density. |
| Carbon sequestration | The carbon sequestration of terrestrial ecosystems is calculated using the net ecosystem productivity (NEP). | Based on energy consumption, the demand for carbon sequestration services is calculated and combined with the raster data of population density. |
| Soil conservation | It is estimated by the Universal Soil Loss Equation in comprehensive consideration of the capacity of the block itself to intercept the upstream sediment. | The actual soil erosion amount is adopted as the demand for soil conservation services. |
| Cultural services | Cultural services are calculated based on the evaluation matrix of the supply and demand of ecosystem services [66]. | The cultural service is measured on the basis of the supply and demand matrix of ecosystem services [66]. |

The ecosystem service supply and demand harmony index (ESDH) is used to measure the balance relationship between the supply and demand of ecosystem services. This study adopts the ecosystem service supply–demand ratio to calculate the ESDH and measure the actual conditions of ecosystem services in the YRB. The specific formula is as follows:

$$S = \sum_{i=1}^{n} S_i \tag{5}$$

where $S$ refers to the total supply of ecosystem services, $n$ refers to the number of ecosystem service supply types, and $S_i$ refers to the standardized value of ecosystem service supply types.

$$D = \sum_{i=1}^{n} D_i \tag{6}$$

where $D$ refers to the total demand of ecosystem services, $n$ refers to the number of ecosystem service demand types, and $D_i$ refers to the standardized value of ES demand types.

$$ESDH_n = \frac{S_n - D_n}{(S_{\max} + D_{\max})/2} \tag{7}$$

where $ESDH_n$ is the supply and demand harmony index of the $n$-th ecosystem service, $S_n$ is the supply of the $n$-th ecosystem service, $D_n$ represents the demand for ecosystem services of $n$, $S_{\max}$ refers to the maximum value of the assessed ecosystem service supply in the evaluation area, and $D_{\max}$ is the maximum value of the assessed demand for a certain ecosystem service in the evaluation area.

The comprehensive ESDH integrates the supply and demand of several different ecosystem services to determine the supply–demand status of the overall ecosystem services. It is calculated as an arithmetic average.

$$ESDH = \frac{1}{n} \sum_{i=1}^{n} ESDH_i \tag{8}$$

where $n$ represents the number of ecosystem services and $ESDH_i$ is the supply–demand harmony index of various types of ecosystem services.

### 2.5. Spatial Autocorrelation Analysis

Spatial autocorrelation is a mathematical representation of spatial dependence that describes the correlation between a variable at a location in space and the same variable at its neighboring location and has been widely used in spatial dependence analyses [82–84]. This study aims to evaluate the spatial aggregation pattern of ecosystem health in the YRB by using spatial autocorrelation analysis. Spatial autocorrelation includes global spatial autocorrelation and local spatial autocorrelation. This study evaluates the spatial aggregation pattern of ecosystem health in the YRB by using local Moran's I index. Local Moran's I index can be employed to analyze the correlation between local elements and adjacent units, and the local indicators of spatial association (LISA) cluster map can be obtained through the clustering and outlier analysis tools in ArcGIS. The LISA map can reflect "high–high", "high–low", "low–high", and "low–low" aggregation in local areas as well as the insignificant situations. The evaluation result of the EVORSH framework is used in the spatial autocorrelation analysis, which reveals the aggregation characteristics of ecosystem health in the YRB. The local autocorrelation coefficients can be formulated as:

$$I_{ij} = Q_{\mathrm{i}}^{m} \sum_{j=1}^{n} (W_{ij} Q_j^z); Q_i^m = \frac{y_i^m - y_m}{\sigma_m}; Q_j^z = \frac{y_j^z - y_z}{\sigma_z} \tag{9}$$

where $I_{ij}$ represents the local spatial autocorrelation coefficient and the $I_{ij}$ value was calculated by using Geoda software to obtain the LISA map; $n$ represents the number of grid cells; $W_{ij}$ represents the spatial weight; $y_i^m$ and $y_j^z$ represent the $m$ and $z$ attribute values of the $i$ and $j$ grid cells, respectively; $y_m$ and $y_z$ represent the average values of attributes $m$ and $z$, respectively; and $\sigma_m$ and $\sigma_z$ represent the variances of attributes $m$ and $z$, respectively.

### 2.6. Geographical Detector

Geographical detector is a statistical method to detect spatial differentiation and study the driving factors that affect such differentiation. It is applicable to qualitative

and quantitative data and can be used to explore the interaction between two factors. In addition, it has collinearity immunity and can ignore the correlation between factors [85]. Geographical detector has been widely used in identifying the factors of climate change, ecosystem services, and economic development [86–89]. This study selects 13 influence factors of ecosystem health starting from two angles, namely, nature and human society, by considering the research of Li et al. [17], Zhong et al. [31], and Yang et al. [32]. Among them, the natural factors include the following: three topographic factors (elevation, slope aspect, and slope), three climate factors (temperature, rainfall, and potential evapotranspiration), water coverage rate (water resource factor), and NDVI (vegetation factor). The human society factors include five factors, namely, population density, gross national product (GDP), urbanization rate (proportion of construction land), proportion of cultivated land, and road network density (Figure 3). The data are processed in ArcGIS. Then, 3 km × 3 km grids are selected as evaluation units, which are divided into five categories via the natural breakpoint method and the equal interval method. The corresponding data are subsequently imported into geographical detector, and the results are analyzed. In this study, differentiation and factor detection and interaction detection in geographical detector are used to explore the main factors that influence ecosystem health.

Differentiation and factor detection are used to measure the spatial differentiation of ecosystem health. The calculated $q$ value is applied to measure the degree of influence of a single independent variable on the differentiation of dependent variables. For a certain factor, when the $q$ value is large, this factor has a strong driving force on the change in the spatial pattern of ecosystem health; otherwise, it has a weak driving force. The $q$ value ranges within (0,1). The $p$ value is used to examine the significance level of the factors, namely, the explanatory power, which is employed to reflect the impact of the factors on ecosystem health evaluation. When the $p$ value of a factor is small, the explanatory power of this factor for the evaluation result is high, and vice versa.

$$q = 1 - \frac{1}{N\sigma^2} \sum_{h=1}^{L} N_h \sigma_h^2 \tag{10}$$

where $h = 1,\ldots; L$ refers to the layering of variable $Y$ or factor $X$; $N_h$ and $N$ are the number of samples of layering and the whole area, respectively; and $\sigma_h^2$ and $\sigma^2$ are the variance of layering $h$ and $Y$ values in the whole area, respectively. The range of $q$ is [0, 1]. When the layering is generated by driving factor $X$, if the $q$ value is large, then the explanatory power of driving factor $X$ for attribute $Y$ is strong, and vice versa.

Interaction detection is utilized to measure whether the combined effect of two driving factors is stronger or weaker than that of a single factor. Its interaction results are divided into five types, namely, nonlinear weakening, two-factor strengthening, single factor nonlinear weakening, nonlinear strengthening, and mutual independence. The study of Wang et al. [85] presents the other specific principles. The explanatory power of two different independent variables for the spatial differentiation of watershed ecosystem health as they act simultaneously is evaluated. The types of interaction are as follows: if $q(X1\cap X2) < \min(q(X1), q(X2))$, it shows nonlinear weakening; if $\min(q(X1), q(X2)) < q(X1\cap X2) < \max(q(X1), q(X2))$, it shows single-factor nonlinear weakening; if $q(X1\cap X2) > \max(q(X1), q(X2))$, it shows two-factor strengthening; if $q(X1\cap X2) = q(X1) + q(X2)$, two independent variables are independent; and if $q(X1\cap X2) > q(X1) + q(X2)$, it shows nonlinear strengthening.

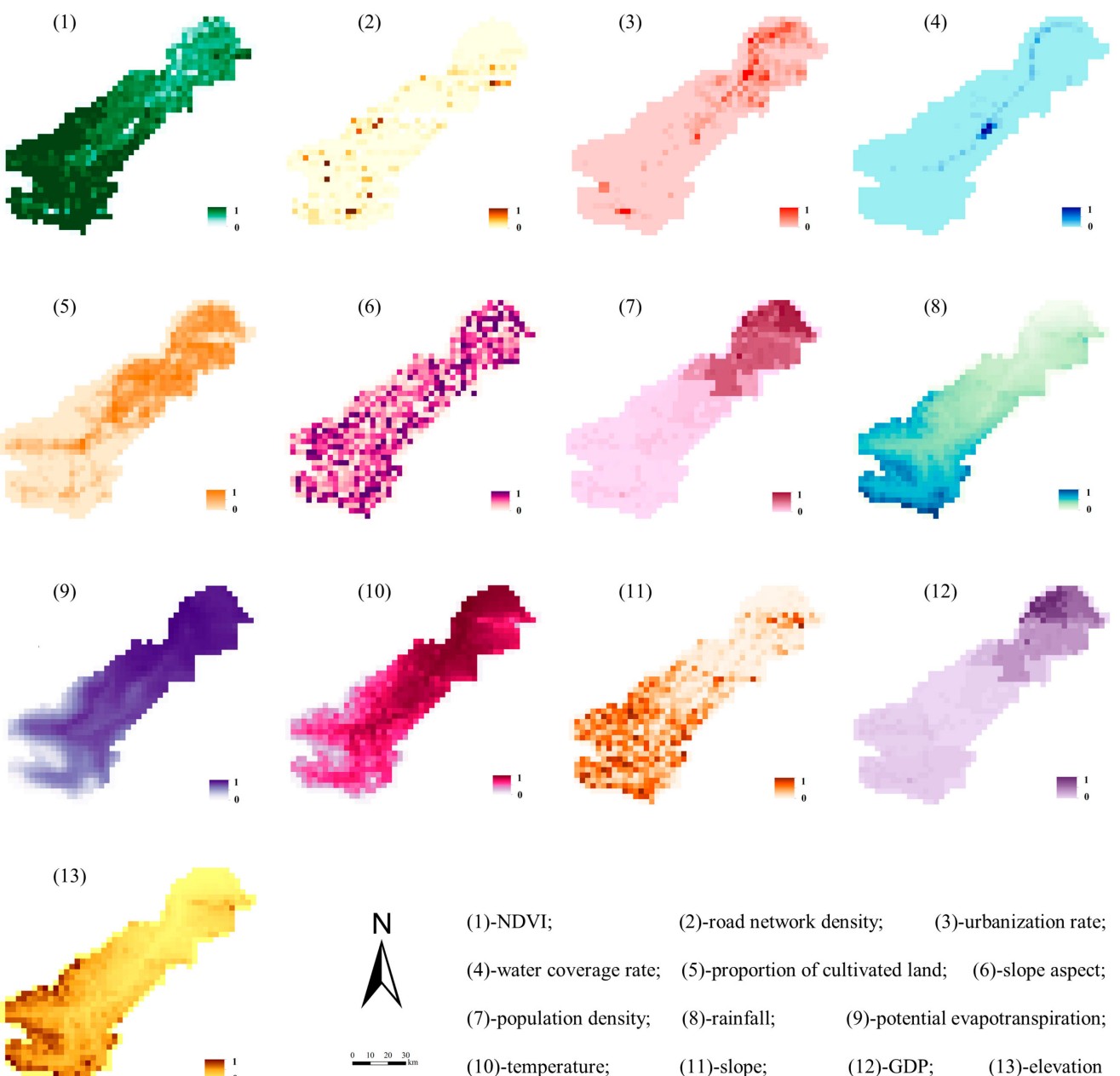

**Figure 3.** Spatial distribution map of restricted factors with 3 km × 3 km resolution.

(1)-NDVI;     (2)-road network density;     (3)-urbanization rate;

(4)-water coverage rate;     (5)-proportion of cultivated land;     (6)-slope aspect;

(7)-population density;     (8)-rainfall;     (9)-potential evapotranspiration;

(10)-temperature;     (11)-slope;     (12)-GDP;     (13)-elevation

## 3. Results

### 3.1. Ecosystem Vigor, Organization, Resilience, and Ecosystem Service Supply–Demand Harmony Index

The spatial expression of ecosystem vigor, organization, and resilience indices was implemented in ArcGIS, and the results are shown on Figure 4. The mean values of ecosystem vigor, organization, and resilience indexes are 0.72, 0.61, and 0.59, respectively, and the areas of the watershed with a value higher than the mean account for 53%, 52%, and 51%, respectively, which are generally half of all grid cells. On the whole, the vigor and organization of the ecosystem are good, and the ecosystem's resilience is average. Ecosystem vigor, organization, and resilience are similar in the overall spatial pattern, showing a decreasing trend from southwest to northeast (Figure 4). The changes in ecosystem organization in terms of spatial distribution are more significant than those in ecosystem vigor and resilience. Ecosystem vigor is characterized by NPP, so Luhun Reservoir, with

a large water area in the middle of the basin, is presented as a low-value-concentration area. However, the elasticity and resistance coefficient of the ecosystem in the water area is relatively high, so the resilience index of the ecosystem in the basin is also high.

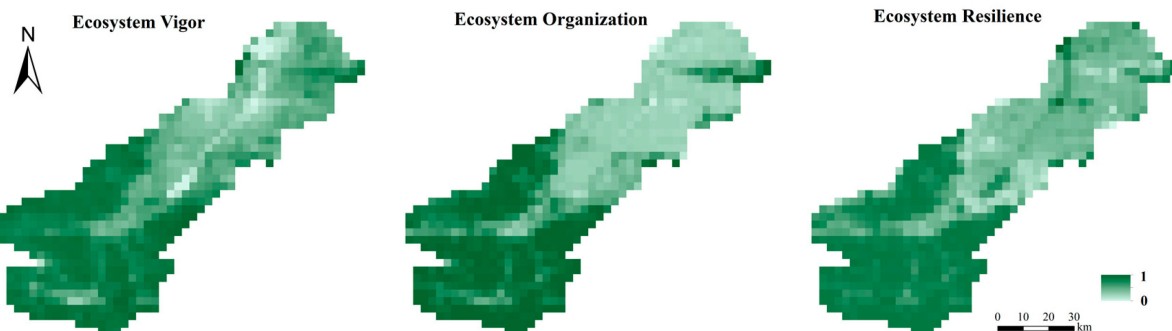

**Figure 4.** Spatial distribution of ecosystem vigor, organization, and resilience in Yihe River Basin.

The mean value of the ESDH is 0.65, and the proportion exceeding the mean value is 53%. The ESDH that is higher than the mean value accounts for about a half. On the whole, the ESDH is at the good level. The supply of ecosystem services decreases from the southwest to the northeast, and the demand for ecosystem services increases from the southwest to the northeast. This finding indicates that the supply and demand of ecosystem services in the YRB do not match in space (Figure 5). The pattern of the ESDH is similar to the supply spatial pattern of ecosystem services. The ESDH in some areas of the upper and middle reaches of the watershed is slightly higher than the supply index, whereas the ESDH in some areas of the lower reaches is slightly lower than the supply index. The ecosystem is not isolated, and human activities affect the ecosystem health in the YRB. Its health level is also relative. The research results show that compared with simply considering the service supply, adding the ESDH can better reflect the actual conditions of local ecosystem services, and it is more in line with the indicators of the ecosystem health assessment (Figure 5).

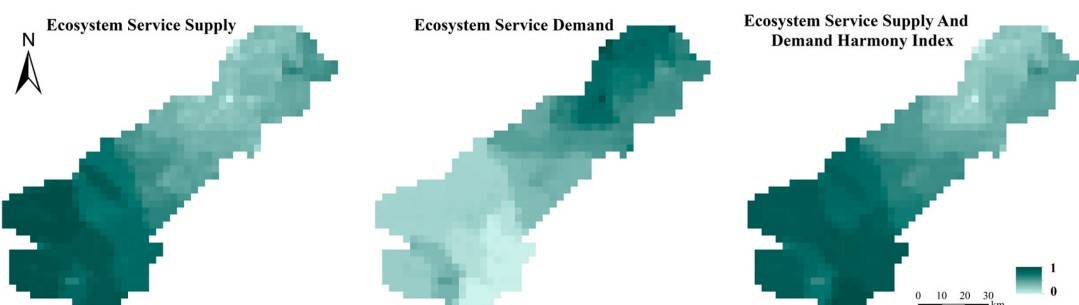

**Figure 5.** Spatial distribution of supply, demand, and supply–demand harmony index of ecosystem services in Yihe River Basin.

### 3.2. Watershed Ecosystem Health Index

The mean value of ecosystem health levels in the YRB is 0.65, and the proportion that exceeds the mean value is 50%. From the perspective of the mean value, the ecosystem health in the YRB is at the good level. The ecosystem health level of the YRB exhibits significant spatial heterogeneity, showing a high spatial pattern in the upper reaches and a low pattern in the middle and lower reaches from the southwest to the northeast. Good and excellent health levels are mainly distributed in the mountain areas of the upper reaches (Figure 6). The local spatial autocorrelation analysis result shows that ecosystem health presents a high–high aggregation distribution in most areas of the upper reaches and a low–low aggregation distribution in a small part of the south of the lower reaches and the

northeast of the middle reaches (Figure 6). In short, the ecosystem health of the YRB shows a decreasing change from the southwest to the northeast. The upper reaches have many areas at the good health level, and the areas at poor and worse health levels are mainly in the lower reaches. The overall results are consistent with the actual situation.

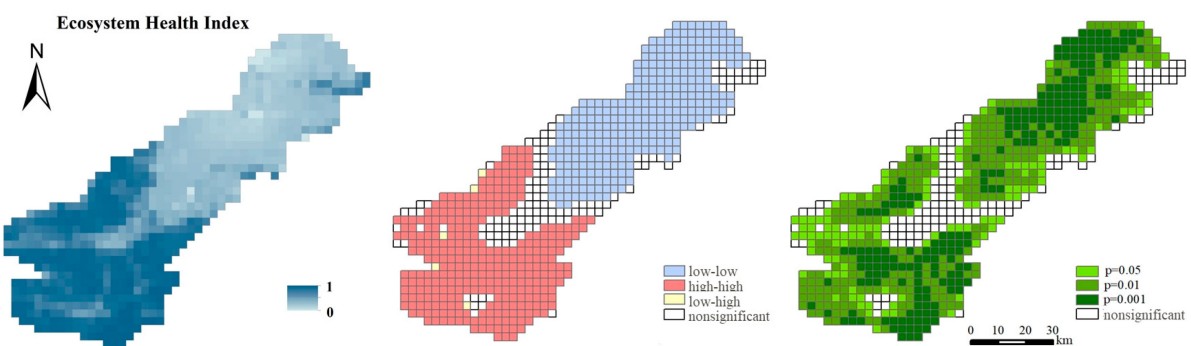

**Figure 6.** Spatial distribution and pattern of ecosystem health in Yihe River Basin.

The areas at the moderate health level account for 24% of the total watershed area; those at the poor health level account for 21%; those at the severe health level account for 1%; and those at the moderate, poor, and severe health levels account for about 50% of the total watershed area. The areas at the excellent health level account for 11% of the total watershed area, and those at the good health level account for 42%. The areas at the good and excellent health levels account for more than 50% of the total watershed area. On the whole, the differentiation of the ecosystem health index in the YRB is significant (Figure 7).

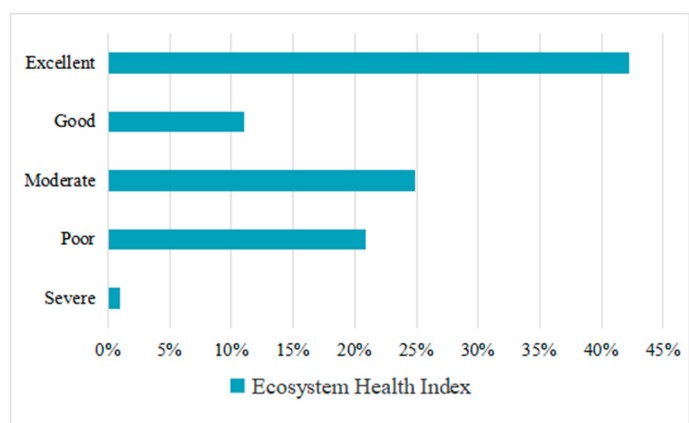

**Figure 7.** Proportions of areas of ecosystem health in Yihe River Basin at all levels.

### 3.3. Influencing Factors of Ecosystem Health

3.3.1. Single-Factor Detection Analysis

The factor detection result from the geographical detector shows that the decisive power ($q$ values) of all factors with regard to the spatial differentiation of watershed ecosystem health level ranks as follows: rainfall > potential evaporation > population density > elevation > GDP > annual temperature > proportion of cultivated land > NDVI > slope > urbanization rate > water coverage rate> slope aspect > road network density (Table 6). Among the 13 influencing factors, population density, potential evapotranspiration, and rainfall have strong explanatory powers of 51.2%, 54.8%, and 55.4%, respectively. The three factors exert the most influence on the spatial differentiation of ecosystem health in the YRB. The effects of factors with explanatory power in the range of 30%–45% on the spatial heterogeneity of ecosystem health increase in the following order: NDVI, proportion of cultivated land, temperature, GDP, and elevation; meanwhile, the explanatory power of the urbanization rate and slope on the spatial heterogeneity of ecosystem health is in the range

of 20%–30%. The $q$ values of road network density, slope aspect, and water area coverage are small, with the $p$ value being higher than 0.05. The effects of the three factors on the spatial differentiation of ecosystem health in the YRB are the smallest, namely, less than 5%. The explanatory power of climate factors, including rainfall, potential evapotranspiration, and temperature, is the strongest. Among the topographic and geomorphic factors, elevation has a more important effect. Among the socio-economic factors, population density and GDP have a relatively important effect. Among the land cover factors, the proportion of cultivated land and NDVI have greater effects on the spatial differentiation of ecosystem health compared with the water coverage rate and urbanization rate.

**Table 6.** Factor detection results of ecosystem health spatial heterogeneity.

| Driving Factors | $q$ | $p$ |
| :---: | :---: | :---: |
| Road network density | 0.003 | 1.000 |
| Slope aspect | 0.011 | 0.174 |
| Water coverage rate | 0.029 | 0.795 |
| Urbanization rate | 0.230 | 0.000 |
| Slope | 0.276 | 0.000 |
| NDVI | 0.343 | 0.000 |
| Proportion of cultivated land | 0.356 | 0.000 |
| Temperature | 0.365 | 0.000 |
| GDP | 0.379 | 0.000 |
| Elevation | 0.426 | 0.000 |
| Population density | 0.512 | 0.000 |
| Potential evapotranspiration | 0.548 | 0.000 |
| Rainfall | 0.554 | 0.000 |

### 3.3.2. Interactive Detection Analysis

As indicated above, the impact of a single factor on the spatial distribution of ecosystem health is analyzed. However, the complex interaction between multiple factors determines the spatial pattern of ecosystem health in practice. According to the interactive detection analysis result (Figure 8), the interaction between any two factors among the 13 categories of influencing factors is significantly enhanced, which is presented in two forms, namely, two-factor enhancement and nonlinear enhancement, without an independent and weakening trend. The impact of the interaction between any two factors on ecosystem health is greater than that of a single factor. In other words, the spatial differentiation of ecosystem health in the YRB is jointly affected by multiple factors. If the value of the interaction is high, then the impact of the interaction between its two corresponding factors on the spatial distribution of ecosystem health is also high. The factors with a strong pairwise interaction effect on the differentiation of ecosystem health are as follows: population density ∩ rainfall (0.63), population density ∩ potential evapotranspiration (0.61), rainfall ∩ potential evapotranspiration (0.60), population density ∩ temperature (0.57), population density ∩ proportion of cultivated land (0.56), population density ∩ GDP (0.58), population density ∩ elevation (0.57), and elevation ∩ potential evapotranspiration (0.59). Population density, potential evapotranspiration, rainfall, temperature, GDP, elevation, and proportion of cultivated land exert a more significant interactive influence on ecosystem health than the other factors. In short, population density, rainfall, and potential evapotranspiration have a great impact on the spatial differentiation of ecosystem health in the YRB. The impact on ecosystem health is more than 50%, and the interaction of the three factors with other factors is highly significant. Hence, water and population are the main factors that affect the spatial differentiation of ecosystem health in the YRB.

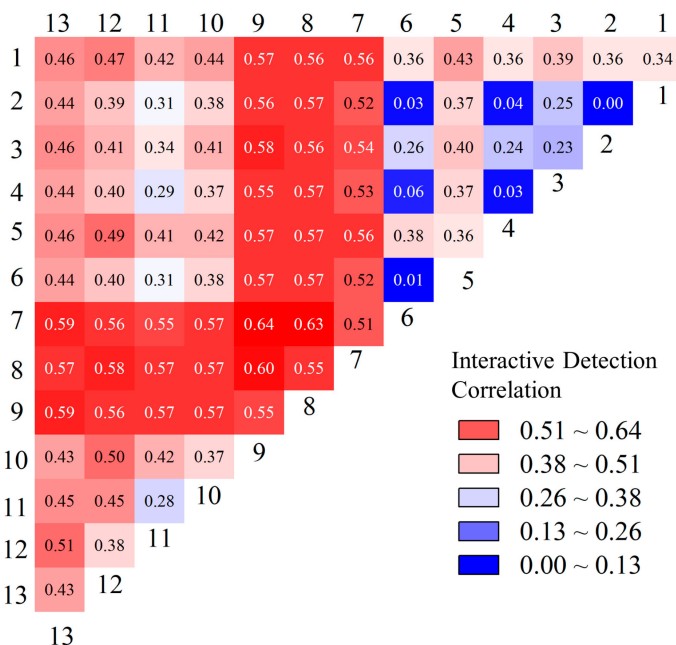

**Figure 8.** Thermodynamic diagram of the interactive detection results. Note: 1-NDVI; 2-road network density; 3-urbanization rate; 4-water coverage rate; 5-proportion of cultivated land; 6-slope aspect; 7-population density; 8-rainfall; 9-potential evapotranspiration; 10-temperature; 11-slope; 12-GDP; 13-elevation.

## 4. Discussion

### 4.1. Watershed Ecosystem Health Assessment Framework

At present, most of the methods used for ecosystem health assessment are based on the establishment of an indicator system. Ecosystem health status is comprehensively evaluated by using the analytic hierarchy process, fuzzy mathematics, and set pair analysis. The advantage of these methods is that an appropriate ecosystem health assessment indicator system is built according to regional characteristics and in full consideration of the diversity and complexity of the factors that affect ecosystem health. However, indicator selection entails a certain subjectivity, and certain rules should be followed. The PSR model mostly relies on the subjective selection of assessment indicators, and ensuring the scientific nature of indicators is difficult. The traditional EVOR model pays particular attention to the integrity and sustainability of the natural ecosystem itself, but it ignores human factors [42]. Reflecting the process of interaction between man and nature is therefore difficult. Accurate ecosystem health assessment is the premise of regional ecological protection and management. The EVOR-based ecosystem health assessment model has been continuously improved. It focused on the health of natural ecosystems in the past. However, it is now widely integrated into ecosystem services to increase the integrity of ecosystems [19,58,60,61,90]. Ecosystem services are closely related to ecosystem health, which is the key to linking ecosystem health and human well-being. Ecosystem services include both supply and demand. However, most scholars have focused on the supply of ecosystem services in their research on ecosystem health, and only a few have considered both [56]. Pan et al. [51] considered the demand of ecosystem services in the establishment of their ecosystem health assessment model. However, the model cannot fully reflect the supply and demand levels of ecosystem services in the selection of ecosystem service supply and demand indicators and lacks the selection of cultural ecosystem service. On the basis of previous studies, the current study includes four categories of supply and demand indices of ecosystem services, namely, provisioning, regulating, supporting, and cultural services, in the ecosystem health assessment. Considering the ecosystem integrity and ecosystem service demand, an ecosystem health assessment framework of EVORSH

is constructed in this work. Compared with the supply of ecosystem services, the ESDH can better represent the actual situation of ecosystem services. The supply and demand coordination of ecosystem services can be effectively integrated into the ecosystem health assessment. With the constructed EVORSH framework, the ecosystem health in the YRB is assessed in this study. The assessment result is basically consistent with the actual situation in the YRB. The constructed ecosystem health assessment framework is an enrichment and extension of the traditional framework, as reflected in the introduction of the ESDH to the ecosystem health assessment. The ESDH is measured based on the supply and demand of ecosystem services, and it reflects the pressure of human society on ecosystem health. In addition, after assessing watershed ecosystem health by using our study framework, the ecosystem health assessment results are comparatively analyzed, and the factors that influence the spatial differentiation and pattern of ecosystem health are explained. On the whole, the constructed assessment framework is feasible at the basin scale. It can provide a theoretical reference for ecosystem health assessment in other basins and a scientific basis for the ecological protection of the YRB and the formulation of macroscopic policies and high-quality development plans. Ecosystem health focuses on natural ecosystem health, considers the supply of ecosystem services, and integrates the demand for ecosystem services due to the impact of human activities, indicating that the coupling relationship between ecosystem health and human society is becoming increasingly close [19].

The assessment results on watershed ecosystem health obtained by three models are compared in Figure 9. The three models are EVOR, EVORS, and EVORSH. After increasing the demand of ecosystem services and introducing the ESDH, the ecosystem health assessment can better reflect the actual situation of the YRB compared with the traditional EVOR and the EVORS, which only consider the supply of ecosystem services. For example, the proportion of areas with poor ecosystem health is increased slightly, and the increased area is mainly in the urban area of Yanshi City, reflecting the negative impact of urbanization on ecosystem health. The areas with poor ecosystem health account for 21%, which is less than 24% of the EVOR and 26% of the EVORS. The ESDH in the decreased areas is mostly at the moderate level (Figure 5), so the assessment result based on the framework built in this study shows that the ecosystem health in these areas is also at the moderate level, resulting in an increase in the area with moderate ecosystem health. The proportion of areas with good ecosystem health is also increased slightly. The reason is that the natural ecosystem in most of the areas in the upper reaches of the YRB face little pressure, and the ESDH is high. The comparison of the assessment results from the three models indicates that the ecosystem health assessment result from the model built in this study is more suitable for the actual situation of the YRB. Therefore, the YRB ecosystem health assessment result is more accurate. The EVORSH framework constructed in this study has a better index system and a more accurate assessment result than the traditional EVOR model and EVORS.

*4.2. Ecosystem Health Assessment at the Grid Scale*

Studies on ecosystem health assessment have focused on regional ecosystems, and in many studies, the units are divided according to administrative regions. However, macro assessment units are inappropriate for small study areas or watersheds with few county-level units. The reason is that the detailed characteristics of the research object will be ignored. In this study, a 3 km × 3 km grid is used as the basic research unit to explore ecosystem health in the YRB; the selected grid unit is more advantageous than macro assessment units. Assessments at the grid scale can make data highly concrete and express the spatial heterogeneity of ecosystem health in the basin. The assessment result is also highly accurate and scientific. On the basis of the constructed assessment framework, this study also assesses the ecosystem health level of the YRB at the sub-basin scale (Figure 10). The assessment result at the sub-basin scale shows the overall ecosystem health level of the sub-basin. The sub-basin assessment averages the ecosystem health state in the sub-basin, neutralizes the data difference in units, and fails to identify the health state inside the

sub-basin. In the assessment result at the grid scale, the high- and low-value partitions of ecosystem health are highly obvious, and the spatial heterogeneity is strong. The grid-scale assessment realizes a fine diagnosis of ecosystem health and can provide location guidance for the targeted solution of punctiform ecological problems. The assessment result at the sub-basin scale shows seven sub-basins (Nos. 3, 4, 6–8, 14, and 26) with ecosystem health at the poor level, accounting for 27% of the total watershed area. This assessment result is inconsistent with the actual ecological environment of the YRB. It further indicates that the ecosystem health assessment result at the grid scale is highly accurate.

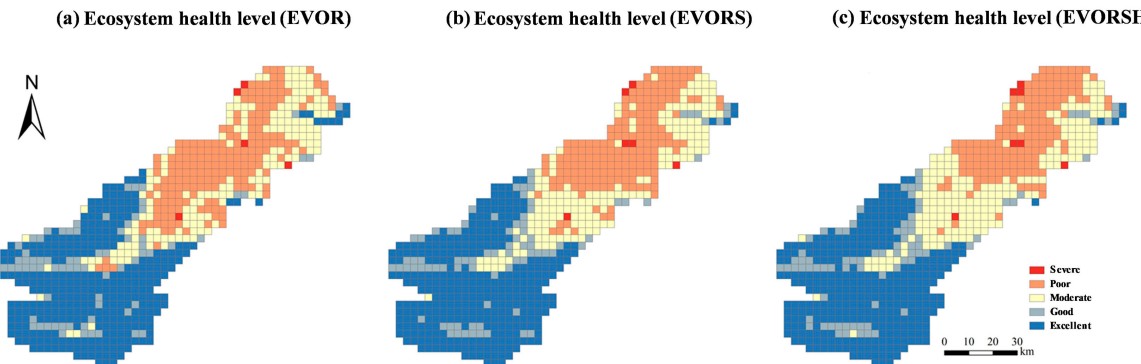

**Figure 9.** Comparison of ecosystem health assessment results. Note: (**a**) is the distribution map of ecosystem health assessed by the traditional EVOR model; (**b**) is the distribution map of ecosystem health assessed by the EVORS model; (**c**) is the distribution map of ecosystem health based on the assessment framework built in this study.

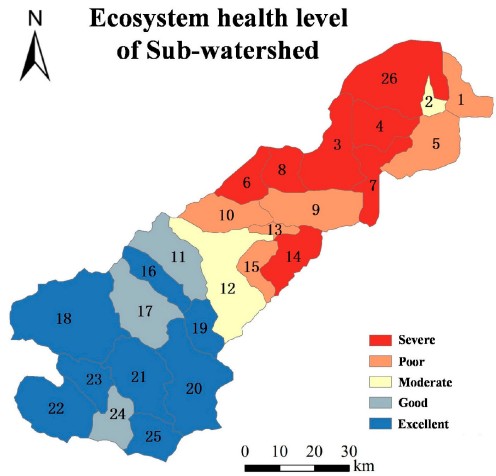

**Figure 10.** Spatial distribution of ecosystem health in Yihe River Basin at the sub-basin (i.e., 1–26) scale.

### 4.3. Factors That Influence Ecosystem Health

Through a systematic analysis of the change characteristics of ecosystem health, the main factors that influence ecosystem health are identified to provide a theoretical basis for managers to formulate policies. Ecosystem health is affected by many factors, including natural and human social factors. Existing studies have achieved some progress in the exploration of the factors that influence ecosystem health. Cao et al. built an ecosystem health indicator system to diagnose the ecosystem health of the Yangtze River Economic Belt and found that population density has the largest weight among all the influencing factors [91]. By using geographical detector, Ran et al. [92] found that the explanatory ability of rainfall with regard to changes in global ecosystem health reaches 57.4%, which

is the primary reason for global ecosystem health differentiation. He et al. [76] reported that significant regional differences exist in the driving forces of ecosystem health in China. In the whole country, the moisture index and land use intensity are the main factors that affect the regional heterogeneity of ecosystem health. In addition, the interaction between the two can enhance the ability of moisture to determine regional ecosystem health differences. Xu et al. assessed the ecosystem health of the Xiong'an urban wetland by using the PSR framework and found that population density has the highest weight among the factors that affect the ecosystem health of the Xiong'an urban wetland [39]. Li et al. [17] assessed the ecosystem health of Xinjiang on the basis of the VOR model and found that NDVI is the primary driving factor that affects the ecosystem health of Xinjiang. Ning et al. [34] constructed a multiple-index system based on the PSR framework to assess the ecosystem health of the Beijing–Tianjin–Hebei region and discovered that the soil factor has the largest weight among the factors that affect the ecosystem health of the Beijing–Tianjin–Hebei region. Yao et al. [90] constructed a model based on the development level–service function–resistance to interference–self maintenance framework to assess the ecosystem health level in various towns and districts of Dongguan. They found that the main factors that contribute to ecosystem health in the various towns and districts include GDP, population growth rate, forest coverage, and water coverage. Some studies have also found that the urbanization level is an important factor that influences ecosystem health in the coastal zone of the East China Sea, the middle reaches of the Yangtze River, Chongqing, and other regions [36,51,93]. Through a bibliometric study, Sun et al. discovered that the factors that affect river ecosystem health include human activities, land use, climate change, and the physical habitat of the river [94].

Among 13 influencing factors determined via geographical detector in the current study, population density, potential evapotranspiration, and rainfall exert the most influence on the spatial differentiation of ecosystem health in the YRB. This result indicates that population and water resources are the main factors that influence the spatial differentiation of ecosystem health in the YRB. The study's conclusion is consistent with the findings of Cao et al. [91], Ran et al. [92], He et al. [76], and Xu Ye et al. [39]. High population density increases the demand for ecosystem services, thereby bringing pressure on the natural ecosystem and affecting the ecosystem health. If high population aggregation exceeds the bearing capacity of the resources and environment, the ecosystem will deteriorate, thereby affecting the ecosystem's health. The areas with high ecosystem health are in the upper reaches of the YRB, where the amount of rainfall is relatively high, the moisture condition and vegetation growth are relatively good, and the supply of ecosystem services is high. Therefore, the moisture factor has a considerable impact on ecosystem health in the YRB. GDP and urbanization level also affect the spatial heterogeneity of ecosystem health in the YRB. Areas with high GDP and urbanization level attract population aggregation, thereby affecting the local ecosystem. Meanwhile, NPP and ecosystem vigor are low in the areas with high urbanization, such as Yanshi City. Urbanization destroys the connectivity of ecological patches, makes the patches fragmented, and results in an islanding effect, which reduces the connectivity of the ecosystem structure and affects ecosystem organization. A certain correlation exists between the influencing factors selected in this study. For example, the meteorological factors, such as rainfall, temperature, and potential evapotranspiration, are complex, and the socio-economic factors, including population, GDP, and urbanization, are related. However, geographical detector has collinear immunity [85], which can effectively exclude the interference of other factors, thereby making the research result highly accurate. Interactive detection is conducted for the 13 influencing factors obtained with geographical detector. The result shows that the combined effect of population density and rainfall can enhance the explanatory ability with regard to the spatial heterogeneity of watershed ecosystem health. Ran et al. [92] and He et al. [76] observed a similar phenomenon. The meteorological and socio-economic factors exert synergistic effects on ecosystem health changes and can be mutually enhanced.

*4.4. Policy Implications*

As for policy recommendations, ecosystem health presents strong spatial heterogeneity, so the ecosystem health management of the YRB should be divided into three sections: upper, middle, and lower reaches for zoning protection, governance, and management. The relationship between the supply and demand of ecosystem services is obviously mismatched in space. For the purpose of regional development and balance, corresponding ecological compensation mechanisms can be established locally. By taking Yihe and the land where the basin is located as a unified whole, in consideration of the integrity of the ecosystem structure and the demand for ecosystem services, the government can manage and control as a whole, regionally coordinate and integrate, maintain the overall structure of the ecosystem, and attach importance to ecosystem services in order to continuously improve the river ecosystem health. Attention should be given to Yanshi City and other regions with low ecosystem health in the lower reaches, and appropriate development and construction should be carried out. For a certain period in the future, the total scale of construction land and the expansion of construction land should be strictly controlled, especially in the central urban area. In addition, forest lands, wetlands, and rivers in the upper and middle reaches, and urban green space in the lower reaches, must be protected and managed. The protection of forest lands and water areas should be enhanced, and the balance point of population, the economy, resources, and the environment should be understood. Population size should also be controlled within the scope of the resources and environment.

## 5. Conclusions

On the basis of the EVOR and EVORS models, this study uses the ESDH, composed of provisioning, regulating, supporting, and cultural services, to construct the ecosystem health assessment system EVORSH for the YRB. Spatial autocorrelation analysis and the geographical detector model are employed to identify the spatial pattern and differentiation factors of ecosystem health in the YRB. Moreover, the watershed ecosystem health assessment framework, assessment scale, and policy implications are discussed. The results show the following:

(1) The ecosystem health in the YRB exhibits significant spatial heterogeneity. High-level areas are mainly distributed in the mountains in the upper reaches of the YRB, and low-level areas are mainly distributed in the plain areas in the lower reaches.

(2) Population density, rainfall, and potential evapotranspiration are the most important driving factors that lead to the differentiation of ecosystem health. This finding indicates that the water resource conditions in the YRB greatly restrict ecosystem health.

(3) The EVORSH framework is suitable for the measurement of ecosystem health in the YRB. The assessment result is consistent with the actual situation in the YRB. A 3 km × 3 km grid is used as the basic research unit, and it can more accurately and scientifically express the spatial heterogeneity of ecosystem health in the YRB compared with the macro evaluation unit.

**Author Contributions:** Funding acquisition, H.W. and L.L.; investigation, Q.H.; methodology, H.W., Q.H. and M.L.; software, Q.H.; visualization, Q.H.; writing—original draft, H.W.; writing—review and editing, Y.Y. and L.L. All authors have read and agreed to the published version of the manuscript.

**Funding:** This research was funded by the National Key R&D Program of China (2021YFD1700900), the National Natural Science Foundation of China (41901259), and the special fund for top talents in Henan Agricultural University (30500425).

**Data Availability Statement:** Not applicable.

**Conflicts of Interest:** The authors declare no conflict of interest.

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
