# Peer review of "Spatial Heterogeneity of Watershed Ecosystem Health and Identification of Its Influencing Factors in a Mountain–Hill–Plain Region, Henan Province, China"

_remotesensing, doi:10.3390/rs15153751_

Round 1
Reviewer 1 Report
The research aimed to assess the watershed ecosystem health and detect the driving factors affecting the health in the Yihe River Basin, China. The manuscript was well-constructed and prepared. The commonly used methods resulted in convincing conclusions. I would suggest acceptance for publication if the authors could make a solid revision by clarifying the novelty of this study. The present manuscript suffered from providing some novel approaches or some new findings except those results focusing on local interests.
Author Response
- Comment:(the authors could make a solid revision by clarifying the novelty of this study. The present manuscript suffered from providing some novel approaches or some new findings except those results focusing on local interests.)
Response: Done as required. Thank you for the suggestions. We have revised the statements in the Abstract section and added some new findings, not just focusing on local interests. “By integrating multi-dimensional data and methods, EVORSH framework proposed in this study can quickly and scientifically assess the status of ecosystem health, identify the influencing factors of spatial heterogeneity, and could be applied in other similar watersheds.” The added conclusion also helps to clarify the novelty of this study. Moreover, the novelty of this study has been stated in the fourth paragraph of Introduction section. “Based on the previous research on ecosystem health, two knowledge gaps still exist in current research. First, scholars cannot reach a consensus regarding the subjectivity and scientific nature of the selection and number of indicators in the research process by the ecosystem health model. The selection of indicators is not well-rounded enough, and the indicator embodiment on the interaction between humans and nature is relatively weak. The ecosystem health assessment model can still be improved in terms of the index system composition. Further exploration is needed to find appropriate evaluation indicators. Second, most ecosystem health assessment frameworks focus on the measurement of the ecosystem health level. The factor interpretation and spatial differentiation mechanism of ecosystem health assessment results are still unclear.”
Reviewer 2 Report
The manuscript presents an analysis of ecosystem health in the YRB, Henan Province, China, the topic is worthy of research. In the current state of the manuscript, it is well written and structured. The manuscript has minimal details.
General comments
C1. Authors could also emphasize particular strengths of the study for potential applications of their method in other regions.
*The answer to these question should be reflected in the manuscript.*
Specific comments
Line 65: Please mention some of these standards.
Line 121-123: And what did you find? What was or were the main finding(s)?
Line 208-Figure 1: Add items a, b, c, for each of the map components.
Line 210: Change "LUCC" to "LULCC".
Line 234: Where (software) was the model implemented or used?
Line 244: What other software, please mention.
Line 243-244: Please add website, add the accessed-on time as day month year
Line 252: What does the difference in colors indicate?
Line 434-439: Why these factors, it is important to support them with relevant literature.
Line 453: Change "[0,1]" to "(0,1)".
Line 509: It is important to add to the graph (Y axis), the legends "severe, by, etc".
Line 544: It is important to add to the graph (Y axis), the legends of "severe, by, etc".
Lin 699. Edit.
Author Response
0.Comment:(Authors could also emphasize particular strengths of the study for potential applications of their method in other regions..)
Response: Done as required. We have added some results, not just focusing on local. “By integrating multi-dimensional data and methods, EVORSH framework proposed in this study can quickly and scientifically assess the status of ecosystem health, identify the influencing factors of spatial heterogeneity, and could be applied in other similar watersheds.”
1.Comment:(Line 65: Please mention some of these standards.)
Response: Done as required. The standards have been added, in the original text “practical standards mentioned primary productivity, nutrients, species diversity, instability, disease prevalence, size spectrum, contaminants.”
2.Comment:(Line 121-123: And what did you find? What was or were the main finding(s)?)
Response: Done as required. We have added the main conclusion to the original article “and found that overall ecosystem health showed a decreasing trend over the two decades 1995-2015, better reflecting the ecosystem health characteristics of the region.”
3.Comment:(Line 208-Figure 1: Add items a, b, c, for each of the map components.)
Response: Done as required. We have added "(a), (b), (c)" to the figure 1.
4.Comment:(Line 210: Change "LUCC" to "LULCC".)
Response: Done as required. We have changed "LUCC" to "LULCC".
5.Comment:(Line 234: Where (software) was the model implemented or used?.)
Response: Done as required. The geographical detector model is programmed in Excel.
6.Comment:(Line 244: What other software, please mention.)
Response: Done as required. We have added Excel software in the text.
7.Comment:(Line 243-244: Please add website, add the accessed-on time as day month year)
Response: ArcGIS10.2, FRAGSTATS4.2, and Excel2016 software are used to process the multi-source data. Software are both stand-alone versions and don’t need the internet.
8.Comment:(Line 252: What does the difference in colors indicate?)
Response: The color difference is for aesthetic purposes, so that the naked eye can better distinguish the construction of indicators, the technical process, and the final goal (Three colors).
9.Comment:(Line 434-439: Why these factors, it is important to support them with relevant literature.)
Response: Done as required. We have added supporting documents to the text, which shows “This study selects 13 influence factors of ecosystem health starting from two angles, namely, nature and human society by considering the research of Li et al. [17], Zhong et al. [31], and Yang et al. [32]. Among them, the natural factors include the following: three topographic factors (elevation, slope aspect, and slope), three climate factors (temperature, rainfall, and potential evapotranspiration), water coverage rate (water resource factor), and NDVI (vegetation factor). The human society factors include five factors, namely, population density, gross national product (GDP), urbanization rate (proportion of construction land), proportion of cultivated land, and road network density (Figure 3).”
10.Comment:(Line 453: Change "[0,1]" to "(0,1)".)
Response: Done as required. We have changed "[0,1]" to "(0,1)".
11.Comment:(Line 509: It is important to add to the graph (Y axis), the legends "severe, by, etc".)
Response: Done as required. In this paper, we only perform "severe, by, etc" for ecosystem health levels, and do not define ecosystem vitality, organization, and resilience, so only the range values are displayed in the figure.
12.Comment:(Line 544: It is important to add to the graph (Y axis), the legends of "severe, by, etc".)
Response: Done as required. In this paper, we only perform "severe, by, etc" for ecosystem health levels, and do not define supply–demand harmony index of ecosystem services, so only the range values are displayed in the figure.
13.Comment:(Edit.)
Response: Done as required.
Reviewer 3 Report
This paper documents development of an ecological metric. The justifications for the way it was constructed should be strengthened to show that the simple relationships used are appropriate. Consider more quantification.
Line 181: consider rewording this sentence.
Line 187: There may be a better comparison-word than "contradiction".
Line 233: What autocorrelation method was used?
Line 239-242: Give a reference justifying the selections of response variables.
Line 245: reword, eliminate "harmonious".
Line 246: State why was the geometric mean used.
Line 253: A thought: This could be recast as a Bayesian Network problem.
Line 268: It would be helpful to show the histograms of the various factors to convince the reader that one variable might dominate this equation.
Line "...Poor..." need to reword to not use "worse" here
Line 282
Line 282: This is useful, but a histogram of the data would be nice.
Line 287... Here, carbon gain by plants is often used as a measure of productivity and light is limited to PAR.
Line 297: these quantities seem important enough to the paper that brief descriptions might be included.
Line 316-317: I'm not sure you want to include such a strong statement of importance here.
Line 337: A reference at the end of this sentence would be useful.
Line 343: reword this sentence.
Table 4: add line breaks to keep parenthetic expressions on a single line. Right-align reasons and convert to bullets or use indentation to separate entries
line 386: Are you sure the components should be equally weighted?
Line 398: The arithmetic average is good for unskewed variables, is that the case here?
Line 411: Showing semivariograms might be instructive
Line 433: If you propose to decompose a single factor into 13 components, a more quantitative description of the methodology is warranted. Are you sure this isn't reversed?
Line 480: "in" a table, "on" a figure
Line 493: was the elasticity coefficient defined earlier in the paper?
Figure 5, maybe 7&9: standard error bars should be added to each bar shown
Figure 10: consider sorting by column (and commensurate row) to group variables by intensity of correlation. What metric is shown, Pearson or Spearman correlation? Wouldn't it make sense to put both on opposite sides of the diagonal trace of the matrix? Also, either type of correlation matrix could be subjected to PCA to illustrate how unique the contribution of each factor is.
Line 699: change red text to black
Author Response
1.Comment:(Line 181: consider rewording this sentence.)
Response: Done as required. We have revised the original text to “Moreover, the increasingly prominent mismatch between the supply and demand for ecosystem services has led to the deterioration of ecosystem health.”
2.Comment:(Line 187: There may be a better comparison-word than "contradiction".)
Response: Done as required. We replace “contradiction” with “mismatch”.
3.Comment:(Line 233: What autocorrelation method was used?)
Response: Done as required. We use the local Moran index in GeoDa software for spatial autocorrelation analysis, which has been explained in the corresponding position in the paper.
4.Comment:(Line 239-242: Give a reference justifying the selections of response variables.)
Response: Done as required. We have added the corresponding reference at the end of the sentence.
- Comment: (Line 245: reword, eliminate "harmonious".)
Response: Done as required.
6.Comment:(Line 246: State why was the geometric mean used.)
Response: Done as required. We have added the corresponding reference at the end of the sentence. Ecosystem vigor, ecosystem organization, ecosystem resilience, and ecosystem service are all essential for ecosystem health, so the geometric mean was used.
7.Comment:(Line 253: A thought: This could be recast as a Bayesian Network problem.)
Response: Thank you for the advice. Figure 2 is the technical process of research framework, including the construction of indicators, the technical process, and the final goal. We learned Bayesian networks are mainly used for inference and prediction, and can be used to calculate the probability distribution of other variables. This network structure allows us to reason based on known information and process incomplete or uncertain data. However, according the existing references, I am not sure whether Bayesian Network can be applied in the field of ecosystem health. Still, thanks for the advice.
8.Comment:(Line 268: It would be helpful to show the histograms of the various factors to convince the reader that one variable might dominate this equation.)
Response: Done as required. The corresponding reference of the equation were added to enhance the convincing. Spatial distributions of the various factors were show in Figure 4, Figure 6 and Figure 8. The histograms are listed in attachment.
9.Comment:(Line "...Poor..." need to reword to not use "worse" here)
Response: Done as required. The word “worse” has been replaced by “weak”.
10.Comment:(Line 282: This is useful, but a histogram of the data would be nice. )
Response: Done as required. Spatial distributions of the various factors were show in Figure 4, Figure 6 and Figure 8. The histograms have been listed in the above.
11.Comment:(Line 287... Here, carbon gain by plants is often used as a measure of productivity and light is limited to PAR.)
Response: Done as required. We have added the brief descriptions in the revised manuscript and as follows. “In the process of NPP, photosynthetically-active radiation (PAR) and actual light energy utilization are key variables. PAR is calculated by the total solar radiation and the fraction of PAR absorbed by the vegetation canopy. The actual light energy utilization is calculated by the growth-limited factors, including temperature and moisture stress on plant photosynthesis.”
12.Comment:(Line 297: these quantities seem important enough to the paper that brief descriptions might be included.)
Response: Done as required. We have added the brief descriptions in the revised manuscript and as follows. “In the process of NPP, photosynthetically-active radiation (PAR) and actual light energy utilization are key variables. PAR is calculated by the total solar radiation and the fraction of PAR absorbed by the vegetation canopy. The actual light energy utilization is calculated by the growth-limited factors, including temperature and moisture stress on plant photosynthesis.”
13.Comment:(Line 316-317: I'm not sure you want to include such a strong statement of importance here.)
Response: Done as required. The statement has been revised. The weight of water bodies and forest patch is set to 0.15.
14.Comment:(Line 337: A reference at the end of this sentence would be useful.)
Response: Done as required. Some references have been added in the revised manuscript.
15.Comment:(Line 343: reword this sentence.)
Response: Done as required. We have condensed the paragraph and removed the sentence.
16.Comment:(Table 4: add line breaks to keep parenthetic expressions on a single line. Right-align reasons and convert to bullets or use indentation to separate entries)
Response: Done as required. We have revised Table 4 as required.
17.Comment:(line 386: Are you sure the components should be equally weighted?)
Response: Thanks for the reminding. The weight of each ecosystem service is equal, which has been appeared in the previous studies [17, 77].
18.Comment:(Line 398: The arithmetic average is good for unskewed variables, is that the case here?)
Response: Thanks for the reminding. One of ecosystem services can be zero, but the comprehensive ecosystem service level should not be affected. Therefore, the arithmetic average not the geometric mean was used.
19.Comment:(Line 411: Showing semivariograms might be instructive)
Response: Done as required. The formula has been added in the revised manuscript.
20.Comment:(Line 433: If you propose to decompose a single factor into 13 components, a more quantitative description of the methodology is warranted. Are you sure this isn't reversed?)
Response: Sorry for the confuse. We did not mean to decompose a single factor into 13 components. The statements have been revised as follows. “This study selects 13 influence factors of ecosystem health starting from two angles, namely, nature and human society by considering the research of Li et al. [17], Zhong et al. [31], and Yang et al. [32].” Geographical detector was applied to identify the influence factors of ecosystem health.
21.Comment:(Line 480: "in" a table, "on" a figure)
Response: Done as required. The word has been revised as required.
22.Comment:(Line 493: was the elasticity coefficient defined earlier in the paper?)
Response: Done as required. We have covered this in section 2.4.3 in the revised manuscript. “The elasticity is the ability to restore the original state of the ecosystem after serious damage.”
23.Comment:(Figure 5, maybe 7&9: standard error bars should be added to each bar shown)
Response: Thanks for the suggestion. Because the evaluation value of each indicator is one and only. We have tried to add the standard error bars in the figures, however, we consider the standard error bars have no meaning in Figure 5, 7 and 9.
24.Comment:(Figure 10: consider sorting by column (and commensurate row) to group variables by intensity of correlation. What metric is shown, Pearson or Spearman correlation? Wouldn't it make sense to put both on opposite sides of the diagonal trace of the matrix? Also, either type of correlation matrix could be subjected to PCA to illustrate how unique the contribution of each factor is.)
Response: Sorry for the confuse. The metric is shown neither Pearson nor Spearman correlation. It is the result of interaction detection between the two factors in the geographic detector, which is introduced in 2.6 of the article. Interaction detection is utilized to measure whether the combined effect of two driving factors is stronger or weaker than that of a single factor. Its interaction results are divided into five types, namely, nonlinear weakening, two-factor strengthening, single factor nonlinear weakening, nonlinear strengthening, and mutual independence. The study of Wang et al. [85] presents the other specific principles. The explanatory power of two different independent variables for the spatial differentiation of watershed ecosystem health as they act simultaneously is evaluated. The types of interaction are as follows: if q(X1∩X2) < min(q(X1), q(X2)), it shows nonlinear weakening; if min(q(X1), q(X2)) < q(X1∩X2) < max(q(X1), q(X2)), it shows single-factor nonlinear weakening; if q(X1∩X2) > max(q(X1), q(X2)), it shows two-factor strengthening; if q(X1∩X2) = q(X1) + q(X2), two independent variables are independent; and if q(X1∩X2) > q(X1) + q(X2), it shows nonlinear strengthening.
25.Comment:(Line 699: change red text to black)
Response: Done as required. We have changed red text to black.

Reviewer 4 Report
This manuscript contains interesting information on the spatial heterogeneity of watershed ecosystem health in a mountain hill plain in China. The following are some specific comments to consider.
Title: Be specific. It is better to mention the country name in the title. Because “typical mountain hill plain” is a vague phrase. I cannot understand how the mentioned area is a typical mountain hill plain.
Abstracts: Results are too general. Please add results with some values.
Line 63: “In 1941, land was connected with health [11]”. The sentence is not clear
Introduction: Very lengthy and contains much information. To me, some of the information is unnecessary and can be removed. Please be concise. Also, the introduction section does not have a proper layout. Please mention objectives clearly.
Line 201: It is not good to add just one value for the temperature. Please add the temperature range.
Lines 243-244: Please add the versions of this software.
Line 278: Please add the reference to the table caption.
In methodology, there is unnecessary information/ description. Please remove them and be concise.
Table 5: In the last column, the authors described the quantification method of ecosystem service demand. If authors have references, please add them here without a detailed explanation.
One more suggestion is moving extra information to either the “Appendix” or “Supplementary Materials”.
In the results section, authors unnecessarily described results which are not needed. Please focus on key points.
Line 630-631: “How to improve the ecosystem health assessment system is the premise of ecosystem health assessment?” This question is not necessary.
Conclusions: The key findings linked to the objectives should be mentioned here. No need additional information.
With the extra information, this looks like a technical report to me. Please be concise and remove unnecessary information. There are several grammatical errors which need to be corrected.
Author Response
1.Comment:(Title: Be specific. It is better to mention the country name in the title. Because “typical mountain hill plain” is a vague phrase. I cannot understand how the mentioned area is a typical mountain hill plain.)
Response: Done as required. The title has been revised as “Spatial Heterogeneity of Watershed Ecosystem Health and Identification of Its Influencing Factors in a mountain–hill–plain region, Henan Province, China”.
2.Comment:(Abstracts: Results are too general. Please add results with some values.)
Response: Done as required.
3.Comment:(Line 63: “In 1941, land was connected with health [11]”. The sentence is not clear.)
Response: Done as required. We have amended it to "In 1941, the concept of land was linked to the notion of health".
4.Comment:(Introduction: Very lengthy and contains much information. To me, some of the information is unnecessary and can be removed. Please be concise. Also, the introduction section does not have a proper layout. Please mention objectives clearly.)
Response: Done as required. Introduction section has been condensed and the unnecessary information has been removed. The length reduces by a fifth. The layout of Introduction section has been revised. The first paragraph is the concept and the meaning of ecosystem health. The second paragraph is the review of evaluation methods. The third paragraph is the review of VOR. The fourth paragraph is focusing on the knowledge gap. The last paragraph is introducing the research contents.
5.Comment:(Line 201: It is not good to add just one value for the temperature. Please add the temperature range.)
Response: Done as required. The statements have been revised as “The annual average temperature and annual rainfall are about 13–15 °C and 600 mm, respectively.”
6.Comment:(Lines 243-244: Please add the versions of this software.)
Response: Done as required. We have added the corresponding version of each software in the text.
7.Comment:(Line 278: Please add the reference to the table caption.)
Response: Done as required. We have added the corresponding reference to the table caption.
8.Comment:(In methodology, there is unnecessary information/ description. Please remove them and be concise.)
Response: Done as required. Methodology section has been condensed and the unnecessary information has been removed. The length reduces by a sixth.
9.Comment:(Table 5: In the last column, the authors described the quantification method of ecosystem service demand. If authors have references, please add them here without a detailed explanation. One more suggestion is moving extra information to either the “Appendix” or “Supplementary Materials”.)
Response: Done as required. The statements on quantification method of ecosystem service demand have been condensed in Table 5.
11.Comment:(In the results section, authors unnecessarily described results which are not needed. Please focus on key points.)
Response: Done as required. Results section has been condensed and the unnecessary information has been removed. The length reduces by a fourth.
12.Comment:(Line 630-631: “How to improve the ecosystem health assessment system is the premise of ecosystem health assessment?” This question is not necessary.)
Response: Done as required. We have removed this sentence “How to improve the ecosystem health assessment system is the premise of ecosystem health assessment?”
13.Comment:(Conclusions: The key findings linked to the objectives should be mentioned here. No need additional information.)
Response: Done as required. Conclusions section has been condensed and the additional information has been removed. The revised conclusions are as follows:
(1) The ecosystem health in YRB exhibits significant spatial heterogeneity. High-level areas are mainly distributed in the mountains in the upper reaches of YRB, and low-level areas are mainly distributed in the plain areas in the lower reaches.
(2) Population density, rainfall, and potential evapotranspiration are the most important driving factors that lead to the differentiation of ecosystem health. This finding indicates that the water resource conditions in YRB greatly restrict ecosystem health.
(3) EVORSH framework is suitable for the measurement of ecosystem health in YRB. The assessment result is consistent with the actual situation in YRB. A 3 km × 3 km grid is used as the basic research unit, and it can more accurately and scientifically express the spatial heterogeneity of ecosystem health in YRB compared with the macro evaluation unit.
14.Comment:(With the extra information, this looks like a technical report to me. Please be concise and remove unnecessary information. There are several grammatical errors which need to be corrected.)
Response: Done as required. Introduction section, Methodology section, Results section and Conclusions section has been condensed and the unnecessary information has been removed. The length of Introduction section reduces by a fifth. The length of Methodology section reduces by a sixth. The length of Results section reduces by a fourth.
Round 2
Reviewer 1 Report
Given the authors have addressed the concerns and improved the manuscript significantly, I would suggest accepting it for publication.
Reviewer 4 Report
The authors addressed the comments and improved the manuscript.